# Alkali-deficiency driven charged out-of-phase boundaries for giant electromechanical response

Haijun Wu [1,2,7✉], Shoucong Ning [2,7], Moaz Waqar [2,7], Huajun Liu [3], Yang Zhang[4], Hong-Hui Wu [5✉], Ning Li[2], Yuan Wu[5], Kui Yao [3], Turab Lookman [6], Xiangdong Ding[1], Jun Sun[1], John Wang [2✉] & Stephen J. Pennycook [2✉]

Traditional strategies for improving piezoelectric properties have focused on phase boundary engineering through complex chemical alloying and phase control. Although they have been successfully employed in bulk materials, they have not been effective in thin films due to the severe deterioration in epitaxy, which is critical to film properties. Contending with the opposing effects of alloying and epitaxy in thin films has been a long-standing issue. Herein we demonstrate a new strategy in alkali niobate epitaxial films, utilizing alkali vacancies without alloying to form nanopillars enclosed with out-of-phase boundaries that can give rise to a giant electromechanical response. Both atomically resolved polarization mapping and phase field simulations show that the boundaries are strained and charged, manifesting as head-head and tail-tail polarization bound charges. Such charged boundaries produce a giant local depolarization field, which facilitates a steady polarization rotation between the matrix and nanopillars. The local elastic strain and charge manipulation at out-of-phase boundaries, demonstrated here, can be used as an effective pathway to obtain large electromechanical response with good temperature stability in similar perovskite oxides.

---

[1] State Key Laboratory for Mechanical Behavior of Materials, Xi'an Jiaotong University, Xi'an, China. [2] Department of Materials Science and Engineering, National University of Singapore, Singapore, Singapore. [3] Institute of Materials Research and Engineering, A*STAR (Agency for Science, Technology and Research), Singapore, Singapore. [4] Instrumental Analysis Center of Xi'an Jiaotong University, Xi'an Jiaotong University, Xi'an, China. [5] Beijing Advanced Innovation Center for Materials Genome Engineering, State Key Laboratory for Advanced Metals and Materials, University of Science and Technology Beijing, Beijing, China. [6] 818 Bishops Lodge Road, Santa Fe, NM, USA. [7] These authors contributed equally:: Haijun Wu, Shoucong Ning and Moaz Waqar. ✉email: wuhaijunnavy@xjtu.edu.cn; wuhonghui@ustb.edu.cn; msewangj@nus.edu.sg; stevepennycook@gmail.com

The properties of a material are strongly correlated with both the "perfect" crystal structures and structural "imperfections". Making appropriate beneficial crystal defects allows one to obtain superior properties over those predicted by "perfect" structures. This has been a strategy of turning imperfections into benefit[1,2]. With precision structural controls at varying levels: micro, meso, nano, and even down to the atomic-scale structural defects, grains, precipitates, interfaces, stacking faults, and dislocations have been widely employed to optimize materials properties. However, previous studies have mainly focused on regular nano-features and higher length scales. The ability to tailor structural defects at the unit-cell scale provides the potential to overcome such a limitation. Point defects exist in many forms, and although innately zero dimensional (0D)[3], they tend to form energetically stable ordered arrays, which can enhance coupling and influence performance[4–6]. Here we employ this effect to demonstrate an ultra-high piezoelectric response.

Lead-based piezoelectric materials, such as Lead Zirconium Titanate (PZT), have been widely used for electromechanical sensors and actuators due to their excellent piezoelectric properties[7]. Environmental concerns over the toxicity of PZT have recently driven the search for lead-free piezoelectric materials. Sodium potassium niobate (KNN) is the most promising lead-free piezoelectric system, owing to its excellent comprehensive performance (high piezoelectric $d_{33}$ and high $T_c$). Since 2004[8], significant progress has been made in KNN based bulk piezoceramics[9–15]. The main strategy has been to construct multiphase boundaries by tuning the chemical compositions so that nanoscale domains with local structural and polar heterogeneity are formed[16–19]. As piezoelectric devices continue to be miniaturized, thin films with high piezoelectric response are in demand for micro- and nano-electromechanical systems (MEMS and NMES) applications[20]. In contrast to the rapid progress in lead-free bulk ceramics, the development of lead-free piezoelectric films lags far behind. Transferring the success of lead-free piezoelectric bulk ceramics into thin films is very challenging due to the severely deteriorated film quality because of alloying and precision control of phases. In addition, high Na/K volatile loss, the difficulty in controlling the complex, highly sensitive chemical dopants[13], and the substrate clamping effect[21], all set further limitations on the synthesis and performance of quality lead-free piezoelectric thin films. Therefore, it is a great challenge to overcome the above limitations and improve the performance of lead-free piezoelectric thin films.

Very recently, we have demonstrated a giant electromechanical response in sodium niobate thin films with extensive alkali vacancies that are likely playing a key role in determining the electromechanical behavior[22]. Herein, we investigate the mechanism in which 0D vacancies drive the formation of 3D nanopillars with 2D strained and charged out-of-phase (OOP) boundaries in lead-free alkali niobate thin films. By tailoring the point defects, we demonstrate the origin of the disruptive change in key electromechanical properties. This new structural strategy of tuning the alkali vacancies, without alloying so that there is no deterioration of film epitaxy, gives rise to a giant electromechanical response in alkali niobate films. In particular, nanopillars enclosed with strained and charged OOP boundaries are formed. Indeed, alkali vacancies are unavoidable in alkali niobate films due to the ease of alkali evaporation during the high-temperature film fabrication. They are commonly known to be detrimental to performance, and thus most previous studies focused on how to suppress or compensate for the evaporation. Our strategy is like "turning waste into treasure", by unconventionally increasing the concentration of the unfavorable alkali vacancies. Accordingly, the excess Nb atoms would occupy the alkali vacancies to form antisite defects, as shown in Fig. 1a, b,

since the formation energy of antisite defects $Nb_{Na}$ is much smaller than that of Na vacancies, especially under Na-deficient conditions, Fig. 1f. For the layer-by-layer growth of the films, antisite defects tend to form 2D $NbO_2$ OOP domains in the first original NaO layer at the film-substrate interface, Fig. 1c1, c2. During the following film growth, 3D nanopillars then form, Fig. 1d1, d2. Such nanopillars possess the same crystal structure as the matrix but share 2D OOP boundaries. Furthermore, the OOP boundaries comprise two neighboring $NbO_2$ layers, which display a different valence from that of the matrix, and thus the boundaries are charged, Fig. 1e. The charged OOP boundaries then act as metastable states for polarization rotation between the nanopillars and the surrounding matrix. Following this strategy, we had previously demonstrated a giant effective piezoelectric coefficient $d_{33}$~1100 pm/V with a high Curie temperature of ~450 °C in NNO films of ~200 nm in thickness[22,23]. Here we demonstrate how a large electromechanical response with good temperature stability can be obtained.

## Results and discussions

Epitaxial NNO films with nanopillars (NP-NNO) were grown on Nb-doped $SrTiO_3$ (001) by a sputtering process (see Supplementary Methods). Aberration-corrected scanning transmission electron microscopy (STEM) was performed to reveal the structural features. The STEM ABF (annular bright-field) in Fig. 2a1–a3 present the overall structural views of NP-NNO/STO films with thicknesses of ~1.6, 23, and 138 nm, respectively, obtained by controlling the sputtering time. A high density of columnar nanopillars are observed to exist generally in all films with large Na deficiency, even for the ultra-thin film with 3–5 unit cells. Fig. 2b is a high-magnification STEM HAADF (high-angle annular dark-field) image illustrating an overall view of the NP-NNO/STO film shown in Fig. 2a2. This shows four nanopillars grown within the film vertically from the interface to the surface. The atomically resolved STEM HAADF and ABF images in Fig. 2c–e and Fig. S1 focus on the boundary between a nanopillar and the matrix, as well as the NP-NNO/STO interface. Clearly resolved are the nanopillar structures with extra atom columns, which project out from the Nb atoms of the overlapped antisite nanopillar and the matrix. Moreover, such extra atom columns present an obvious shift ($\delta_C$) along the out-of-plane c direction. Considering the Z contrast of STEM imaging, the extra atom columns are clearly visible in the HAADF image, and they are more obvious than Na and O atoms in the ABF image, which suggests that they are Nb atoms.

The boundary between the nanopillar and the matrix is fully coherent and vertically sharp, from the interface to the surface. The film-substrate interface is also fully coherent, i.e., an ideal epitaxy is established, which would benefit the film performance. From STEM imaging, the termination of the STO substrate is resolved as $TiO_2$ in this region. Accordingly, the first layer of the film should be the NaO layer. However, since Na is quite deficient at the beginning, some Nb atoms could occupy Na positions as $Nb_{Na}$ antisites and then form a 2D region with OOP boundaries. This can be inferred from the abnormally bright contrast of the first NaO layer, compared with the film matrix, as shown in Fig. S2. Once the 2D OOP region forms, the film grown on top of it forms a nanopillar. The nanopillar shares vertical OOP boundaries with the matrix. In other regions, where the first layer does not possess antisite Nb atoms, the film becomes the matrix. Therefore, these antisite Nb atoms in the first layer determine the subsequent growth pattern of nanopillars with OOP boundaries. The STO substrates used in this work are mixed $TiO_2$ and SrO terminations. For the case of SrO terminations, the first layer $NbO_2$ of the film is grown as normal, but nanopillars can still be

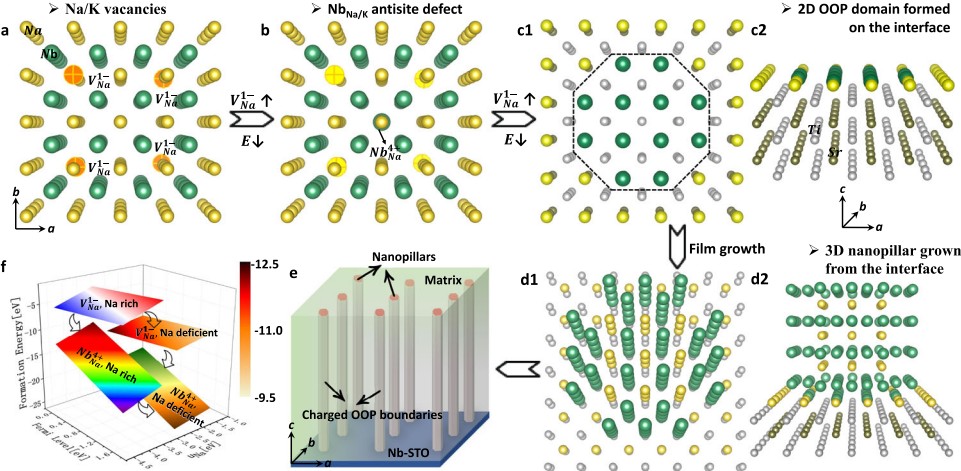

**Fig. 1 Strategy for the formation of nanopillars with charged OOP boundaries for giant effective piezoelectricity in alkali niobate films. a, b** Atomic models showing 0D Na vacancies and one $Nb_{Na}$ antisite defect replacing the vacancies. **c1, c2** Two views of the atomic model showing the interface with 2D OOP domain grown on the STO substrate for the first layer of the film. **d1, d2** Two views of the atomic model showing a 3D nanopillar grown on STO substrate for the following film growth. **e** Schematic for nanopillars with charged OOP boundaries embedded in the matrix. **f** Calculated defect formation energies for Na vacancies and $Nb_{Na}$ antisite defect, as a function of Na chemical potential and Fermi level. O atoms are not shown in the structural models for simplicity.

formed from the second layer, NaO. In the regions with both SrO and $TiO_2$ terminations, nanopillars prefer to form on the $TiO_2$ termination, whereas the SrO termination is followed with the film matrix, as shown in Fig. S3.

It is difficult to resolve the detailed structure of nanopillars using only the cross-sectional view, since the thickness of the sample greatly exceeds that of the nanopillars, and nanopillars overlap with the matrix. To reveal the nature of the nanopillars, plan-view STEM was performed so that the nanopillars without the substrate (polished and then ion milled during the sample preparation) could be viewed individually from the top. As expected, a high density of nanopillars with sizes ranging from a few to tens of nanometers is observed, as shown in Fig. 2f, which is consistent with those observed along the cross-sectional direction. It is clear that the nanopillars are exactly the same as the matrix, but with a shift of a half unit cell along both in-plane directions. Consistent with our previous report[22], the relative unit-cell shift $\delta_C$ between nanopillars and the matrix along the c direction, observed along the cross-sectional views, can be described by $<a/2, b/2, \delta_C>$, where a and b are unit vectors along in-plane directions and $\delta_C$ is the displacement along the out-of-plane direction. The relative misfit ($\delta_C$) along the out-of-plane direction clearly differentiates the present boundary from the conventional antiphase boundary. The well-known antiphase boundary, often found in Ruddlesden–Popper type structures[24,25], is characterized by a missing $BO_2$ (B being B-site atom) layer between two regions of the same phase, where one region is shifted by a distance of (a/2, b/2, c/2) from its original position. However, in the present case an AO (A being A-site atom) layer is missing between two perfect perovskite regions with one region shifted by $(a/2, b/2, \delta_C)$ instead of (a/2, b/2, c/2). This is different from an antiphase boundary but still belongs to an OOP boundary, displacing a lattice mismatch of a fraction of a unit cell dimension in neighboring regions of the crystal[22,25]. Thus, we still refer to it as an OOP boundary.

The critical feature of these OOP boundaries is that they are strained and charged, as discussed later, which is at the root of the unusual piezoelectric behavior reported previously. The inset in Fig. 2f and Fig. S4 give the strain analysis of Fig. 2f by means of geometric phase analysis[26,27], showing couples of high positive/negative strains at the boundaries. Fig. 2g–j display enlarged

images of the representative nanopillars with regular shapes containing 6, 8, 12, and 16 Nb atom columns. The smallest nanopillars possess 6 Nb atom columns and display the strongest atom distortion. As shown in Fig. 2g, the 4 Nb atoms (marked with white dashed circles) at the boundaries as well as the 6 Nb atoms inside the nanopillars are strongly distorted, deviated far from their initial positions; accordingly, the O atoms surrounding them are also severely distorted, Fig. S5a2. As the nanopillar size expands, the distortion decreases, as shown in Fig. 2g–j. The boundaries with Nb enrichment form just to compensate for the Na deficiency. The nanopillars with smaller sizes segregate relatively more Nb atoms compared to the larger nanopillars, and thus present much stronger lattice distortion.

To further verify the structure, we have established 3D structural models of the NP-NNO/STO heterogeneous structure to reproduce the experimental observation. Fig. 2k is a 3D structural model with one polar nanopillar surrounded by a matrix grown on the NNO/STO interface. For the initial structural model, the center part of the film was shifted by <a/2, b/2, 0>. The interface is an STO substrate with $TiO_2$ termination connecting the first NaO layer of the NNO film, where Nb atoms occupy the Na positions in the nanopillar region. Fig. 2l1, l2 present views of the initial states of nanopillars along [001] (plan view) and [100] (cross-section view) axes. After the structural relaxation, the optimized structures are shown in Fig. 2m1, m2 and Fig. S6, where the in-plane distortion and out-of-plane relative shift ($\delta_C$) of the whole nanopillar are shown. The in-plane distortion is induced by the two directly connected Nb-O planes at the boundaries, whereas the out-of-plane shift is caused by the abnormal $TiO_2$-$NbO_2$ (B-B) connection instead of the normal $TiO_2$-NaO (B-A) at the interface. Based on structural models, we have performed STEM image simulations for both plan and cross-sectional views. As shown in Fig. 2n, o and Fig. S7, the simulated images are very consistent with the experimental views.

We next consider the valence of Nb at the boundaries. With increasing nanopillar size, the fraction of boundary Nb atoms decreases relative to the whole nanopillar, so the excess concentration of Nb atoms will decrease. To maintain electrical neutrality, Nb enrichment implies a lower Nb valence, especially at the boundaries. As shown in Fig. S8, the average Nb valence of the nanopillar with 6 Nb atoms is the lowest, 3.33 +, increasing

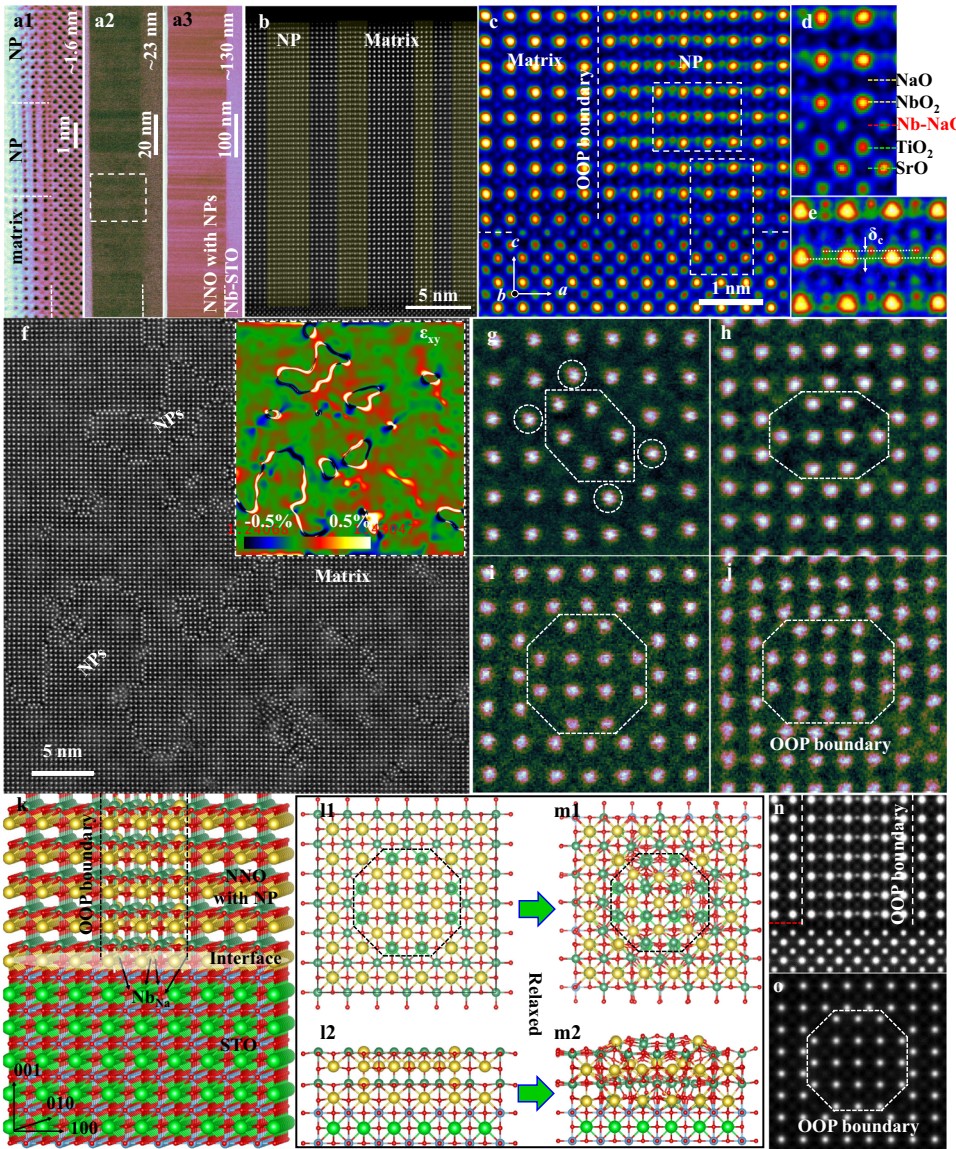

**Fig. 2 Structural imaging of NP-NNO thin films. a1–a3** STEM ABF images of the NP-NNO/STO film grown with thickness of ~4 uc, ~23 nm, and ~130 nm, respectively. **b** High-magnification STEM HAADF image showing an overall view of the NP-NNO/STO film shown in **a2**. **c** Atomically resolved STEM HAADF image showing one OOP boundary and the NNO/STO interface. **d** Enlarged image from **b** focusing on the interface. **e** Enlarged image from **c** focusing on the nanopillar. **f** High-magnification plan-view STEM HAADF image of NP-NNO/STO showing a high density of nanopillars, with a strain analysis result inset. **g–j** Enlarged STEM HAADF images of one nanopillar with 6, 8, 12, 16 Nb atoms, respectively. **k** The 3D structural model with one polar nanopillar and the NNO/STO interface. **l1**, **l2 and m1**, **m2** Initial and relaxed structural model of one polar nanopillar with 12 Nb atoms and the NNO/STO interface, viewed along [001] (plan view) and [100] (cross-section view) axes. **n**, **o** Simulated STEM HAADF images of NP-NNO/STO with one nanopillar.

to 3.75+ for nanopillars with 8, 12, and 16 Nb atoms, and finally approaching 5+ as the size of nanopillars increases to infinity (close to the matrix). This is summarized in Fig. S5. The lowering of Nb valence at the boundaries suggests the existence of charge and charge redistribution to lower the energy, which can dramatically influence the local polarizations of the nanopillars, especially at the boundaries.

It is well known that local spontaneous polarization arises from the electric dipole formed by the relative displacement of the centers of negative and positive ions. For ferroelectric alkali niobates, the displacement of the center polar metal Nb5+ cation with respect to the corner Na+ cations ($\delta_{Nb-Na}$) or face-center O2- anions ($\delta_{Nb-O}$) can be used to represent the local polarization. To accurately locate the column positions, the intensity of each atom column was fitted to a 2D Gaussian peak[28]. Fig. S9a is an atomically-resolved STEM HAADF image of NP-NNO/STO

film, with the nano pillar regions being colored red. Fig. S9b is an arrow map of $\delta_{Nb-Na}$ (i.e., polarization). The polarizations of the nanopillars are almost along the out-of-plane direction, whereas the polarizations of the matrix deviate slightly in the out-of-plane direction. The overall polarization is along the out-of-plane direction and the average displacements are ~39 pm, more than twice that of bulk NNO (~15 pm)[29]. To calculate the relative lattice shift between nanopillars and the matrix, we have calculated the atomic displacements of nanopillar-Nb atoms in the nanopillar region and matrix-O atoms (as a reference) in the matrix region with respect to their nearest matrix-Nb atoms in projection. Fig. S10a, b are a pair of STEM HAADF and contrast-reversed ABF images showing the peaks for the displacement calculation. The atomic shift of the extra nanopillar-Nb columns with respect to the overlapped matrix-Nb columns, $\delta_C$, is ~74 pm, (Fig. S10c), ~13% of the out-of-plane lattice parameter c of NNO.

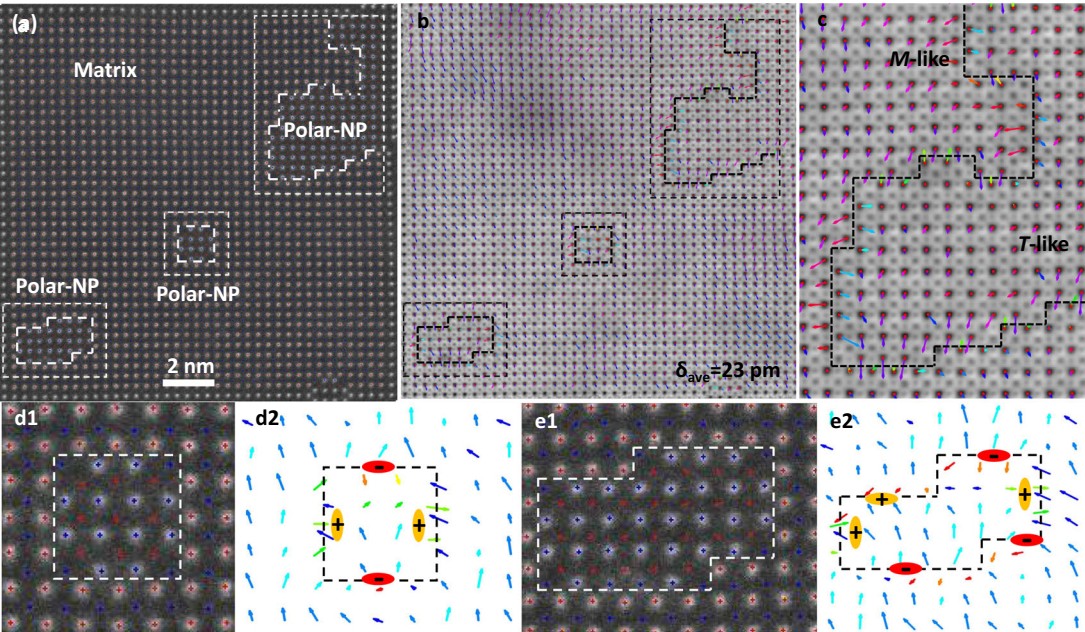

**Fig. 3 Charge and polarization analysis. a** Plan-view STEM HAADF image of NP-NNO film, where three nanopillars are marked. **b** Colorized polarization ($\delta_{Nb-O}$) arrow map overlaid on the respective STEM ABF image of **a**, where the corresponding nanopillars are marked. **c** Enlarged image of the region with a large nanopillar marked in **b**. **d1**, **e1** Enlarged images of the regular nanopillar with 12 Nb atoms and polar nanopillar. **d2**, **e2** Colorized polarization arrow map of **d1**, **e1**, respectively.

Note that this is a displacement, not a direct measure of local polarization since the shift is a projected shift and the nanopillars are much less than the thickness of the specimen. Thus, the shift is not necessarily within one unit cell of the nanopillar.

In a plan view sample, the nanopillars do run through the entire sample thickness. The polarization state along the in-plane direction, especially around the boundaries, can then be directly measured from the observed atom positions. Fig. 3a, b are simultaneously acquired STEM HAADF and ABF images of the NP-NNO film showing three nanopillars. All Nb, Na, and O atom columns of nanopillars and the matrix are found and fitted to 2D Gaussian peaks. Lattice parameter maps along x and y directions are presented in Fig. S11b1, b2, where the high-strain state is seen around the nanopillars. The colorized polarization ($\delta_{Nb-O}$) arrow map is given in Fig. 3c. The overall polarization of both nanopillars and the matrix is roughly along one of the <100> directions, downward as shown in Fig. 3c, except at the boundaries. The preferential in-plane polarization orientation is related to the slightly different a/b parameters. The average displacements are ~23 pm, larger than that of bulk NNO. The nanopillars display much smaller in-plane polarization than the matrix, as shown in Fig. 3c, which suggests that the nanopillars are more tetragonal (T)-like while the matrix maintains a monoclinic (M)-like phase. Fig. 3d1, e1 present enlarged images of one regular nanopillar with 12 Nb atoms and another relatively irregular nanopillar with marked peaks overlaid. Their respective polarization analyses are shown in Fig. 3d2, e2. Interestingly, the boundaries of both nanopillars are charged, displaying one pair of head-to-head (positively charged) polarizations along one a/b while the other pair of tail-to-tail (negatively charged) polarizations along the perpendicular b/a direction. The polarization analysis results are well consistent with the strain analysis, i.e., paired of tail-to-tail and head-to-head polarization states. The high positive strain state possesses high tension, i.e., atoms displacing away, which corresponds to the tail-to-tail polarization state, while the high negative strain corresponds to the head-to-head polarization state. Both the strain analysis and polarization analysis can well

support each other. The charged boundaries are related to the abnormal charges here due to Nb excess or Na deficiency. We also performed STEM differential phase contrast (DPC) imaging to evaluate the local electric field around the nanopillars with charged OOP boundaries. In the STEM-DPC imaging, the transmitted electrons in the bright field of the diffraction plane are collected by multiple detectors, typically four quadrants. The defection of the electron beam, which is in a linear relation with the local electric field, can be measured using the signals recorded by these detectors[30]. The electric field is averaged along the growth direction of nanopillars to account for sample thickness variation and multiple scattering. As shown in Fig. S12, there is an increase in the electric fields in the nanopillar region, which reflects that the OOP boundaries of the nanopillars are charged.

To shed further light on the formation mechanism of the charged boundary structure around the nanopillar, we carried out phase field simulations to study the contributions from misfit strain and bound charge. We simulated the ferroelectric domain structure of NP-NNO films with different conditions at the STO substrate. As discussed in the Supplemental Information, the misfit strain in the nanopillar region is set to be −0.8%, whereas the misfit strain in the matrix is set to be 0.1% along a axis and 0.4% along b axis as the overall structure of the film displays different a and b[22]. As shown in Fig. S13a1, the 3D structure of strained NP-NNO film displays a nonuniform feature with the polarizations in the nanopillar region tending to point along the out-of-plane direction with tetragonal-like symmetry while the matrix maintains monoclinic-like symmetry. The 2D slice images along the middle section of the simulated film are shown in Figs. S13(a2-a4), respectively. If only the strain is considered, the simulated results are not fully consistent with the experimental observations. However, if we consider the bond charge, the simulated images of the model with charge density ±2.0 are shown in Fig. S13b and Fig. 4a, b1–b3. In contrast to the conventional head-to-tail configuration in Figs. S13a1–a4, the preferred orientation of the domain structure around the positively charged OOP boundary is a head-to-head structure. Around the

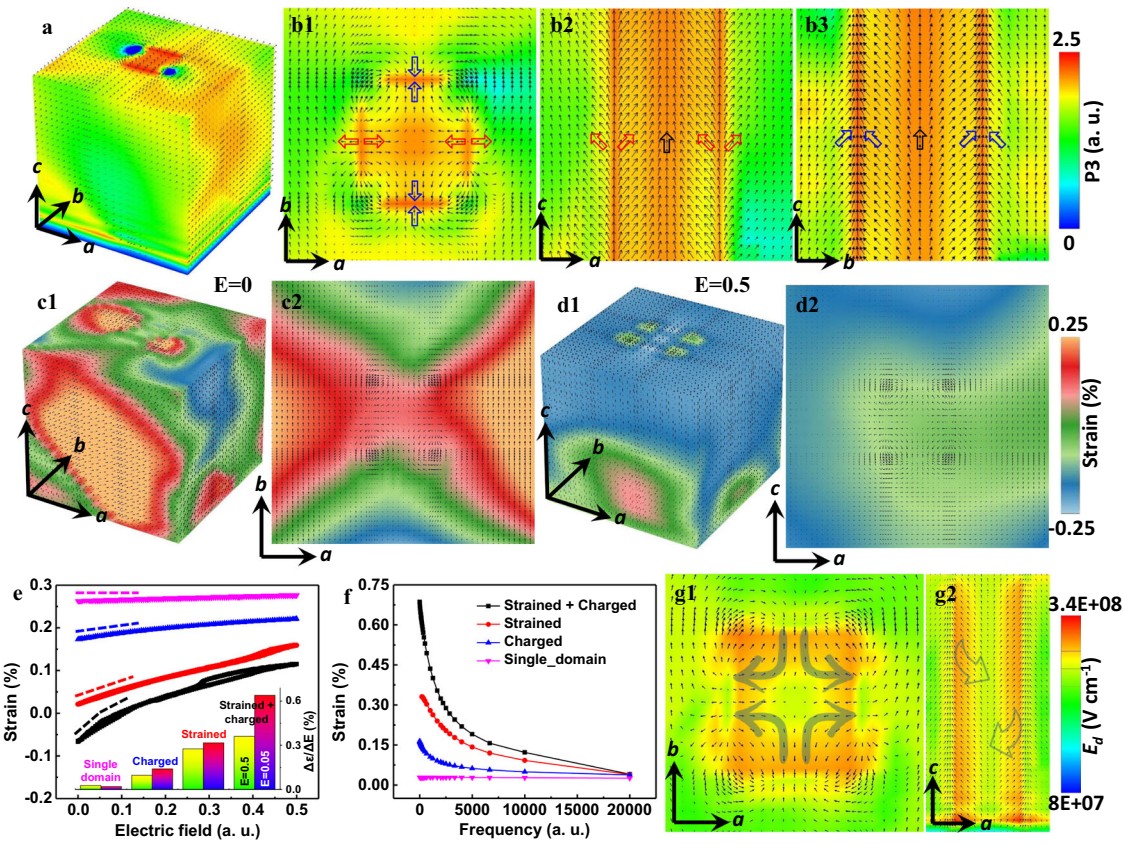

**Fig. 4 Phase field simulations of strained and charged OOP boundaries. a** 3D view of the domain structure in a NNO film with a strained and charged nanopillar. **b1–b3** 2D slice images along *c*, *b*, *a* axes respectively at the middle section of **a**. Regions with different colors indicate ferroelectric domains with different polarization directions. **c1**, **c2** and **d1**, **d2** 3D and 2D strain maps of the strained and charged NP-NNO film at the initial state and under an applied electric field of 0.5 a.u. (arbitrary unit) respectively. **e** Average effective electric-field induced strain of single-domain NNO film for only strained, only charged, and charged + strained NP-NNO films. The frequency is 10 a.u. The inset is the strain change over the electric field range of 0.05 and 0.5. **f** Average effective electric-field induced strain as a function of frequency of the four cases in **e** with an applied electric field of 0.05 a.u. **g1**, **g2** 2D slice images along *c* and *b* axes, respectively showing the depolarization field.

negatively charged OOP boundary, the domain structure is tail-to-tail, as shown in Fig. 4a, b1–b3. This is mainly because the bound charge at the OOP boundary breaks the polarization continuity (i.e., $\nabla \cdot P = -\rho_{BC}$, where $\rho_{BC}$ is the volume density of the bound charge and $\nabla$ is the gradient operator). Figs. S14–S16 show the phase field simulations of different misfit strain conditions in the nanopillar and OOP boundary, as well as different charge conditions. The results show that the misfit strain influences the overall phases of the matrix and the nanopillar, whereas bound charges are critical to forming the head-to-head or tail-to-tail polarizations at the boundaries and significantly affect the polarizations around the boundaries. It is worth noting that the model with all negative or positive charges at boundaries produces a vortex-like polarization distribution inside the nanopillar, as shown in Fig. S17, which isolates the nanopillar from the matrix. If we consider the small lattice difference along a and b of the film, the boundaries of a closed nanopillar with a lower valence of Nb atoms should be nearly equivalent, i.e., it should hold the same charge. However, it is a higher energy state. The true state tends to form pairs of head-to-head or tail-to-tail polarizations with a lower the total energy, which can facilitate the polarization rotation between the nanopillars and the matrix. The driving force for the polarization evolution is the electrostatic and elastic interactions during the energy minimization.

To correlate the strained and charged OOP boundaries with the enhanced electromechanical response, we further calculate the effective electric-field induced strain as functions of applied

electric field and frequency. Figure 4c1, c2 display a 3D and 2D (a-b plane) effective electric-field induced strain maps of the strained and charged NP-NNO film at the initial state (E = 0). The initial state displays strong strain variation derived from the boundaries. With increasing applied electric field, the overall strain approaches a homogeneous distribution (Fig. 4d1, d2 and Fig. S18b1–b4, c1–c4). On unloading the field, the strain state can recover well the initial state, Fig. S18d1–d4, suggesting good recoverability. To evaluate the contribution of lattice strain and bound charges on the enhanced electric-field induced strain response at the boundaries, we simulated the electric-field dependence of average electric-field induced strain. We consider four cases: single-domain NNO film, strained NP-NNO film, charged NP-NNO film as well as charged and strained NP-NNO film. As shown in Fig. 4), the strain curve is reversible during electric-field cycling, which indicates that the response of the polarizations at the boundaries under applied electric field is reversible. Moreover, the behavior of the domain structure with respect to the electric field can be described by the slope of the curve shown in Fig. 4e, that is, by the strain change over the applied electric field range shown in Fig. 4f. This increases steadily from a single-domain NNO film, through a strained or charged NP-NNO film, to a strained and charged NP-NNO film, suggesting that both lattice strain and bond charge contribute to the enhanced electromechanical response significantly. Furthermore, we also simulated the frequency dependence of electric-field induced strain for the four cases by applying an A.C. electric

field shown in Fig. 4g and Fig. S19. The domain boundary response is much stronger upon lowering the frequency for the NP-NNO film. For the single-domain NNO film, the effective electric-field induced strain is almost independent of frequency, as expected. The NP-NNO films with strain, charge, and strain plus charge, show a clear frequency dependence of electric-field induced strain. The strained and charged NP-NNO film showed the strongest frequency dependence consistent with the experimental observations. We conclude that the giant strain response to the electric field originates from the behavior of strained and charged OOP boundaries.

It is well known that charged domain walls can contribute to enhanced electromechanical response[31–33]. Charged domain walls possessing a high density of bound charges require a charge compensation mechanism. The compensation can come from electron transfer across the band gap due to the band-bending mechanism[31], from extrinsic charges related to ionic defects such as oxygen[34], or even cationic vacancies segregated at the domain wall[32,33]. The present case of charged OOP boundaries is a special case of the formation of charged domain walls. Importantly, the boundaries are Nb-rich or Na-deficient with lower Nb valence and thus charged, which could be the extrinsic charge source for the dominant contribution to charge compensation. Theoretical calculations based on the Ginzburg–Landau–Devonshire model shows that the magnitude of the extrinsically introduced charge at the domain wall, and the nanoscale domain size can both promote rotation of the static polarization vector within the body of adjacent domains[32]. Furthermore, as distinct from the traditional non-180° charged walls of the laminate domain structure, e.g., 90° head-to-head/tail-to-tail charged twin walls of $BaTiO_3$, our case presents pairs of 180° charged OOP boundaries. The closed domain boundaries make the depolarization field, $E_{dep}$, inhomogeneous and gradually rotated, which is consistent with the gradual polarizations observed in Fig. 4g1, g2 and Fig. S20. This is different from the homogeneous situation within the traditional laminate domains[31–33]. The magnitude of our calculated $E_{dep}$, inside the nanopillars with closed OOP boundaries, is ~$3.4 \times 10^8$ $Vm^{-1}$ for NNO using $E_{dep} = -P/\varepsilon_0\varepsilon_{st}$[32], where P is the polarization, $\varepsilon_0$ and $\varepsilon_{st}$ are the vacuum permittivity and static dielectric constant of the material, respectively. It is significantly larger than that for conventional micro-scale laminate domains ($E_{dep}$ ~$1.5 \times 10^6$ $Vm^{-1}$) in $BaTiO_3$ using $E_{dep} = E_g$/ed, which depends on the band gap $E_g$ and the domain width $d$[31],. However, it is relatively lower than that for nanodomains free of extrinsic charges in NNO ($E_{dep}$ ~$2 \times 10^9$ $Vm^{-1}$) using the same method. The relatively reduced $E_{dep}$ due to external charges at the boundaries, and its gradual distribution, can stabilize the boundary region into a metastable state. The nanosized domains and charged boundaries with extrinsic charges synergistically contribute to such giant built-in $E_{dep}$.

Based on our phase field calculations, the NP-NNO film has a monoclinic-like symmetry, while the nanopillars with the strained and charged OOP boundaries tend to possess tetragonal-like symmetry. The head-to-head or tail-to-tail polarizations at the charged boundaries can act as a metastable state between them. The most energy-saving path for polarization switching between the two stable states would follow a rotation-like path across the metastable state at the charged boundaries. The local giant built-in $E_{dep}$ could facilitate such polarization rotation, which is a prerequisite for giant piezoelectric response. The dynamic factor responsible for the enhanced piezoelectricity is that polarization rotated by $E_{dep}$ could approach a point of instability. At this point, a small excitation by an external electric field along the c direction is sufficient to induce a large change of polarization along the out-of-plane direction[31].

In summary, we have demonstrated how to manipulate commonly unfavorable 0D A-site defects in alkali-deficient perovskite epitaxial NNO film to drive the formation of 2D charged OOP boundaries and subsequent 3D nanopillars in a thin film. The importance of this is that it can give rise to an effectively large piezoelectric response. The charged boundaries present themselves as head-to-head or tail-to-tail polarizations and produce local built-in depolarizing fields, which are much higher than in usual domain boundaries. These can act as metastable states to facilitate polarization rotation between the matrix and nanopillars, and thus provide a giant electric-field induced strain. Our work represents a departure from the traditional strategy of complex alloying, and sheds light on a promising capability to generate a large response in piezoelectric films which is applicable to other material systems with intrinsic defects.

## Data availability

The authors declare that all data supporting the finding of this study are available within the paper and its Supplementary information files.

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

## Acknowledgements

H. J. Wu would like to acknowledge the financial support from Singapore Ministry of Education Tier 1 grant, R-284-000-212-114, for Lee Kuan Yew Postdoctoral Fellowship and the financial support from the Top Young Talents Programme of Xi'an Jiaotong University. J. Wang and K. Yao acknowledge the supports by A*STAR, under RIE2020 AME Individual Research Grant (IRG) (Grant No.: A20E5c0086), for research conducted in both National University of Singapore and Institute of Materials Research and Engineering (IMRE, A*STAR) Singapore. The authors would like to thank the strong support from Instrumental Analysis Center of Xi'an Jiaotong University. S. J. Pennycook and J. Wang would like to acknowledge the financial support by the Ministry of Education, Singapore under its Tier 2 Grant (Grant No. MOE2017-T2-1-129). H. H. Wu acknowledges the financial support from the Natural Science Foundations of China (No. 51901013 and 52071023).

## Author contributions

H.J.Wu conceived the main idea and led the project. H.L. and M.W. fabricated the NNO thin films. H.J.Wu and Y.Z. performed and analyzed the aberration-corrected STEM experiments. H.H.Wu and H.J.Wu performed the DFT and the phase-field simulations. S.N. and H.J.Wu performed STEM image simulation. H.J.Wu, J.W., S.J.P., H.H.Wu, M. W., K.Y., and H.J.Liu wrote the manuscript with contributions from others. T.L., X.D., J. S., N.L., and Y.W. provided helpful discussion and revised the manuscript. All authors discussed the results and commented on the manuscript.

## Competing interests

The authors declare no competing interests.
