## [Peer Review File · Nature Communications]

REVIEWER COMMENTS

Reviewer #1 (Remarks to the Author):

The authors demonstrated a strategy in alkali niobate epitaxial films, utilizing alkali vacancies to form nanopillars enclosed with out-of-phase boundaries that can give rise to a giant electromechanical response. This work is considered as an extension of the authors' recent work, "Giant piezoelectricity in oxide thin films with nanopillar structure", (Huajun Liu et al, Science, 2020, 369(6501):292-297). In the previous work, the authors also formed nanopillar regions in perovskite oxide thin films to emulate a local heterogeneity, and obtained a giant effective piezoelectric coefficient d_{33} of ~ 1100 pm/V with a high Curie temperature of $\sim 450^\circ\text{C}$. It is difficult to catch the novelty/originality of this work, since the main achievements have been reported.

Also, I have some minor comments for the authors:

[1] Figure 1, especially the atomic models and the view images (A-D), lack clear annotations. The atoms (Na, Nb, and O) and coordinate axis (or crystallographic axis) are necessarily labeled.

[2] The inset in Figure 2F and Figure S3 give the strain analysis of Figure 2F, and high strain states at the boundaries. The authors should provide some explanations about the appearance of plus and minus, viz, under what situations, strain is positive or negative, respectively?

[3] The authors claimed that, "The average displacements are ~ 23 pm, larger than that of bulk NNO." The value for that of bulk NNO or related literature should be provided.

[4] Has any scanning probes (e.g. EFM or KPFM) been used to examine the charged OOP boundaries embedded in the matrix? The authors may be too quick to attribute the observed giant electromechanical response to the strained and charged boundaries.

[5] Some minor mistakes. "As the nanopillar size expands, the distortion decreases, as shown in Figs. 3(G-J)" should be "Figs. 2(G-J)". "(O1, O2) and (M1, M2) Simulated STEM HAADF/ABF images of NP-NNO/STO with one nanopillar." should be "(O1, O2) and (P1, P2)". The axis for Figure 4(C1) is missing.

Reviewer #2 (Remarks to the Author):

Wu and co-authors present in their manuscript a detailed experimental analysis by STEM of the pillar structure of alkali-deficient NaNbO_3 thin films that show a giant electromechanical response. The extensive experimental data are complemented with DFT calculations and phase-field simulations. The authors show that the out-of-phase boundaries separating the pillars from the matrix are strained and charged, and that these effects can explain the extraordinary properties of these special NP-NNO films. The data presented are of highest quality and the analyses and interpretation of the data are convincing. Overall, this work, as a follow-up work of ref. 22 of the manuscript, is of broad interest as it provides detailed structural data that can explain the extraordinary physical properties of these thin films. I do have a few suggestions I would like the authors to address before recommending publication of the work in Nature Communications:

- The most important point concerns the presentation of the data in figures that have numerous subfigures whose presentation is too small to see important details discussed in the text. For instance, p. 5 refers to "abnormally bright contrast of the first NaO layer" in Fig. 2C and 2D: I simply cannot see that abnormally bright contrast. Fig. 2 is simply too dense and should be reduced to show less examples while the actual details should be better visible. Other examples: I cannot see any difference of the atomic positions between Fig. 2L1 and 2M1, except that the bonding sticks become less ordered. Fig. S6a: I don't see any arrows and there is no color scale for the direction of the arrows etc. Although the data are of highest quality, the authors should consider to show less of these data (but clearer) in the main manuscript and add the additional examples in the SI.

- The concept that the boundaries of the pillars are charged is not directly shown. Could the authors use an alternative technique, like EELS or iDPC to provide better experimental evidence of the charged domain walls. What I miss in this context concerns the discussion of possible oxygen vacancies that could balance possible charging effects to the cations. The current experimental data do not unambiguously confirm a charged state the boundaries. The authors show that the boundaries are strained, but it is not shown that they are charged as mentioned on p. 7.
- On p.12 it is explained that the calculations strain curve is reversible. I wonder what effect would need to be taken into account in the calculations to show an actual irreversibility. Could the calculations indeed show an irreversibility?
- I personally believe that it is important to clearly state in the present manuscript what is new compared to reference 22, where already structural models etc. were explained in detail.

Minor

- p.4: ABF does not stand for "angle annular bright-field" (skip angle)
- p.8: comparison between experimental data and STEM simulations: what means "very consistent"? Just by visual inspection or were intensities compared on an absolute scale?
- p.8: reference to Fig. S4 should probably be S5
- p.9: "Figs. 3(D2,D2)"
- p.10: it is referred to x and y axes, but they are nowhere defined.

Referee #1:

General comment: “The authors demonstrated a strategy in alkali niobate epitaxial films, utilizing alkali vacancies to form nanopillars enclosed with out-of-phase boundaries that can give rise to a giant electromechanical response. This work is considered as an extension of the authors’ recent work, “Giant piezoelectricity in oxide thin films with nanopillar structure”, (Huajun Liu et al, *Science*, 2020, 369(6501):292-297). In the previous work, the authors also formed nanopillar regions in perovskite oxide thin films to emulate a local heterogeneity, and obtained a giant effective piezoelectric coefficient d_{33} of ~ 1100 pm/V with a high Curie temperature of $\sim 450^\circ\text{C}$. It is difficult to catch the novelty/originality of this work, since the main achievements have been reported.”

General response: Thank the referee for the evaluation and for his/her valuable suggestions.

In the previous work, we had reported the giant piezoelectricity, and the present work goes much beyond what we reported in the previous *Science* (2020) work. Since the breakthrough work, we have discovered further important findings and developed novelties with lead-free piezoelectric/ferroelectric oxide thin films, in particular in the significantly new scientific understanding and systematic tuning toward the generalized property optimization, which are desperately needed for establishing the on-going topic of lead-free oxide piezoelectric/ferroelectrics.

In our previous *Science* paper, we reported on the unexpectedly giant electromechanical response of ferroelectric thin films of a simple ternary oxide, namely NaNbO_3 . However, there were several key scientific puzzles that were un-solved at that time, for example, (i) how were such self-assembling nanopillars formed exactly? (ii) why were such extended structural features so beneficial to piezoelectricity, i.e., what were the structure-property correlation and the physical origin behind? Without properly addressing and understanding them, it would be impossible to grasp how to further boost the piezoelectric/ferroelectric properties of this material system and other associated systems, and to extend this strategy to other lead-free piezoelectric/ferroelectric systems. As a matter of fact, there is a big family of NaNbO_3 -based piezoelectrics/ferroelectrics.

Therefore, there is an apparent timely need to establish the unique formation mechanism and uncover the underlying structural/physical origins.

In a timely response, in the present work, we have thoroughly explored a new point-defect engineering strategy, where the otherwise detrimental 0D alkali-deficiencies are purposely utilized to drive the formation of 3D nanopillars enclosed with the 2D charged out-of-phase boundaries in alkali niobate epitaxial films. With this new point-defect engineering strategy, the formation of 3D nanopillars is enabled and enclosed with the 2D charged out-of-phase boundaries. By employing atomically resolved polarization mapping and phase field simulations, we show that these boundaries are strained and charged, and they present as the couples of head-head and tail-tail polarization bound charges. Such charged boundaries can produce a giant local depolarization field, which facilitates a steady polarization rotation between the matrix and nanopillars, giving rise to giant electromechanical response.

The novelties and significances of the present work can be briefly summarized as follows:

- We present a brand-new strategy for forming the self-assembled nanodomains, based on point defect ordering in NaNbO_3 thin film.
- We report a brand-new ferroelectric boundary, enclosed and charged out-of-phase boundaries, which present as couples of head-head and tail-tail polarizations, as a new pathway to boost the piezoelectricity.
- We guide new strategies to further enhance the piezoelectricity of NaNbO_3 and even its analogue $(\text{K}, \text{Na})\text{NbO}_3$ thin films, through precisely regulating the Na/K ratio and the deficiency, and the density, shape and even in-plane distribution of the columnar nanopillars.

Actually, with the thorough understandings on the formation mechanism and structural/physical origin of the novel extended structural features, we have managed to precisely manipulate the alkali deficiency as well as the density of the charged boundaries. More recently, we have achieved an even much higher piezoelectric property in a doped- NaNbO_3 epitaxial thin film, $d_{33} \sim 1900$ pm/V, almost double that of the previous *Science* work. With the key scientific understandings being made, the significance of the present work will definitely go much beyond what we know now for the big family of NaNbO_3 and $(\text{K}, \text{Na})\text{NbO}_3$ based

piezoelectrics/ferroelectrics.

Detailed comment 1: “[1] Figure 1, especially the atomic models and the view images (A-D), lack clear annotations. The atoms (Na, Nb, and O) and coordinate axis (or crystallographic axis) are necessarily labeled.”

Response: We have added the labels for Na/Nb/Sr/Ti atoms and coordinate axes. Please see the revised Figure 1, where O atoms are not shown in the structural models for simplicity.

Detailed comment 2: “[2] The inset in Figure 2F and Figure S3 give the strain analysis of Figure 2F, and high strain states at the boundaries. The authors should provide some explanations about the appearance of plus and minus, viz, under what situations, strain is positive or negative, respectively?”

Response: Thank the referee for the valuable suggestion. Yes, we have added some explanations in the revised manuscript, as seen in Page 7 and 10.

For nanopillars enclosed with the out-of-phase boundaries, it is clearly shown that there are couples of high positive/negative strains in the strain analysis, which are well consistent with the paired of tail-to-tail and head-to-head polarization states. The high positive strain state possesses high tension, i.e., atoms displacing away, which corresponds to the tail-to-tail polarization state, while the high negative strain corresponds to the head-to-head polarization state. Both the strain analysis and polarization analysis can well support each other.

Detailed comment 3: “[3] The authors claimed that, “The average displacements are ~ 23 pm, larger than that of bulk NNO.” The value for that of bulk NNO or related literature should be provided.”

Response: We have added the related literature, *Phys. Rev.* 1968, 172, 551-553, in the revised manuscript.

Detailed comment 4: “[4] Has any scanning probes (e.g. EFM or KPFM) been used to examine the charged OOP boundaries embedded in the matrix? The authors may be too quick to attribute the observed giant electromechanical response to the strained and charged boundaries.”

Response:

As one of the scanning probe techniques, aberration-corrected scanning transmission electron microscopy (STEM) provides real space imaging and spectroscopy at atomic resolution with a new level of sensitivity to structure, bonding, elemental valence and even dynamics, thus it is more powerful than other scanning probes (e.g. EFM or KPFM). In the present work, we have confirmed the charged OOP boundaries from three direct techniques, based on aberration-corrected STEM: (i) strain analysis: couples of the high positive/negative strains states, as shown in Figures 2F and S4; (2) atomically-resolved polarization analysis: couples of the tail-to-tail and head-to-head polarization states, as shown in Figure 3; (3) EELS analysis: Nb valence decrease around the boundaries, as shown in Figure R1. They can well support each other and concurrently confirm the charged OOP boundaries.

Moreover, to reveal the correlation between the novel charged OOP boundaries and the giant electromechanical response, we also performed phase-field simulation to evaluate the contribution of strains and bound charges on the polarizations with and without external electric field. The experimentally observed tail-to-tail and head-to-head polarization states around the enclosed out-of-phase boundaries of the nanopillars can be well re-produced after introducing the bound charges into the strained system.

By both experimental studies and theoretical calculation, we can safely conclude that the existence of charged OOP boundaries and the key contribution to giant electromechanical response.

Detailed comment 5: “[5] Some minor mistakes. “As the nanopillar size expands, the distortion decreases, as shown in Figs. 3(G-J)” should be “Figs. 2(G-J)”. “(O1, O2) and (M1, M2) Simulated STEM HAADF/ABF images of NP-NNO/STO with one nanopillar.” should be “(O1, O2) and (P1, P2)”. The axis for Figure 4(C1) is missing.”

Response: We are sorry for the mistakes. We have corrected them in the revised manuscript.

Referee #2:

General comment: “Wu and co-authors present in their manuscript a detailed experimental

analysis by STEM of the pillar structure of alkali-deficient NaNbO₃ thin films that show a giant electromechanical response. The extensive experimental data are complemented with DFT calculations and phase-field simulations. The authors show that the out-of-phase boundaries separating the pillars from the matrix are strained and charged, and that these effects can explain the extraordinary properties of these special NP-NNO films. The data presented are of highest quality and the analyses and interpretation of the data are convincing. Overall, this work, as a follow-up work of ref. 22 of the manuscript, is of broad interest as it provides detailed structural data that can explain the extraordinary physical properties of these thin films. I do have a few suggestions I would like the authors to address before recommending publication of the work in Nature Communications:”

General response: Thank the referee for evaluation and the comment of our work.

Detailed comment 1: “- *The most important point concerns the presentation of the data in figures that have numerous subfigures whose presentation is too small to see important details discussed in the text. For instance,*

p. 5 refers to “abnormally bright contrast of the first NaO layer” in Fig. 2C and 2D: I simply cannot see that abnormally bright contrast.

Fig. 2 is simply too dense and should be reduced to show less examples while the actual details should be better visible.

Other examples: I cannot see any difference of the atomic positions between Fig. 2L1 and 2M1, except that the bonding sticks become less ordered.

Fig. S6a: I don’t see any arrows and there is no color scale for the direction of the arrows etc.

Although the data are of highest quality, the authors should consider to show less of these data (but clearer) in the main manuscript and add the additional examples in the SI.”

Response: Thank the referee for his/her kind suggestions on these figures. We have amended these figures and made them clearer and more concise in the revised manuscript.

We have also added Figure S2 in the revised version, to show “*abnormally bright contrast of the first NaO layer*” from the intensity profile.

We have also simplified Figure 2, and moved some of it into supporting materials (Figures S5 and S7) in the revised version.

For Fig. 2L1 and 2M1, after the long-time structural relaxation, the Nb atoms at the two sides of

the OOP boundary displace away from each other.

We have changed the original Figure S6 to Figure 9, where the arrows can be seen clearly.

Detailed comment 2: “- The concept that the boundaries of the pillars are charged is not directly shown. Could the authors use an alternative technique, like EELS or iDPC to provide better experimental evidence of the charged domain walls. What I miss in this context concerns the discussion of possible oxygen vacancies that could balance possible charging effects to the cations. The current experimental data do not unambiguously confirm a charged state the boundaries. The authors show that the boundaries are strained, but it is not shown that they are charged as mentioned on p. 7.”

Response: We have employed STEM-EELS to evaluate the Nb valence state around the boundary, to be reported in a new manuscript. It clearly shows the valence decrease of Nb atoms around the boundary.

To further ascertain that the lowering of Nb valence occurs at the PF interface, we also used atomically resolved energy-loss near-edge spectroscopy (ELNES) at the O K-edge, Figure R1. The valence of Nb ions was qualitatively obtained from the O K edge using the following method. In the perovskite structure of KNN, oxygen atoms in the octahedron are covalently bonded with the Nb atom where O 2p orbitals are hybridized with 4d orbitals of Nb. [*Microsc. Microanal.* 12, 416–423 (2006)] Each Nb atom is surrounded by oxygen octahedron in perovskite phase which causes the splitting of degenerate 4d orbitals into e_g and t_{2g} orbitals with the former having slightly higher energy. The oxygen K loss edge is characterized by 1s electron transition to 2p unoccupied shell where the peaks a and b (Figure 1Rb) correspond to the t_{2g} and e_g components of the Nb 4d – O 2p orbital overlapping respectively. [*Microsc. Microanal.* 12, 416–423 (2006)] The transition from d_0 configuration in Nb⁵⁺ to d_1 configuration in Nb⁴⁺ (or even lower valence) involves the filling of t_{2g} orbitals, which are lower in energy, hence decreasing the t_{2g}/e_g ratio. This decrease is also reflected in the a/b ratio in the O K edge, hence it is a reliable indication of the valence change of Nb in our case [*Ultramicroscopy* 203, 82–87 (2019)] and shows a partial filling of 4d orbital. Based on the above analysis, we calculated the factor $\gamma = (I_b - I_c)/(I_a - I_c)$ (I is the intensity, a, b and c are the peaks labelled in Figure 1Rb) to qualitatively measure the Nb valence in the vicinity of the OOP boundary which should increase with decrease in Nb valence.

Figure R1. STEM-EELS for the charged OOP boundary. (a) STEM HAADF image around an OOP boundary; (b) the EELS curves obtained from different regions marked in (a). (c) the factor $\gamma = (I_b - I_c)/(I_a - I_c)$ where I is the intensity, a, b and c are the peaks labelled in (B).

As suggested, we performed STEM-DPC imaging to evaluate the local electric field around the nanopillars with charged OOP boundaries. In the STEM-DPC imaging, the transmitted electrons in the bright field of the diffraction plane are collected by multiple detectors, typically four quadrants. The deflection of the electron beam, which is in a linear relation with the local electric field, can be measured using the signals recorded by these detectors. The electric field is averaged along the growth direction of nanopillars to account for sample thickness variation and multiple scattering. As expected, there is an increase in the electric fields in the nanopillar region, as shown in Figure R2, which reflects that the OOP boundaries of the nanopillars are charged. We have added it as Figure S12 into the revised *Supporting Materials*.

Figure R2. STEM-iDPC for the charged OOP boundary. (a) STEM HAADF image showing one nanopillar; **(b)** the corresponding generated distribution of the electric fields, with averaged line profile of the electric field along the horizontal direction.

We agree with the referee, that one shall consider the possible oxygen vacancies to balance (or compensate) the possible charging effects to the cations. For alkali-based oxides, the formation energy of alkali vacancies is much lower than those of oxygen vacancies. From the ELES data, there is no obvious decrease in the total intensity of O K edge around the boundary, as shown in Figure R1b, which reflects that oxygen vacancies are not the major way. In the present work, a large alkali-deficiency drives the formation of Nb-rich charged out-of-phase boundaries, which could well compensate alkali vacancies. The Na deficiency or Nb sufficiency with lowered Nb valence at the OOP boundary could be the extrinsic charge source for the dominant contribution to charge compensation for charged boundaries.

Detailed comment 3: “- On p.12 it is explained that the calculations strain curve is reversible. I wonder what effect would need to be taken into account in the calculations to show an actual irreversibility. Could the calculations indeed show an irreversibility?”

Response: For Figure 4E, we calculated the strain as a function of applied electric field during loading and unloading for cycles and the strain curves are reversible.

Detailed comment 4: “- I personally believe that it is important to clearly state in the present manuscript what is new compared to reference 22, where already structural models etc. were explained in detail.”

Response: This has been addressed in the reply to a similar comment raised by the other referee. In the previous *Science* (2020), we presented a basic schematic for the structural model, where only an interface is demonstrated. By contrast and going much beyond, we have established a super structural model in the present work, on the basis of the new scientific understanding in both the novel structure formation and the physical origins for the giant piezoelectric response. For the nanopillars within the matrix on the STO substrate, we considered and demonstrated the key contributions from strains and charges, which are much closer to the real structure. Based on the super structural cell, we have conducted DFT relaxation and phase-field calculations, and also further simulated the STEM images, as shown in Figure 2(O, P), which can well be consistent with the experimental observation.

Detailed comment 5: “p.4: ABF does not stand for “angle annular bright-field” (skip angle)”

Response: We are sorry for the mistakes. We have corrected it in the revised manuscript.

Detailed comment 6: “p.8: comparison between experimental data and STEM simulations: what means “very consistent”? Just by visual inspection or were intensities compared on an absolute scale?”

Response: To fully simulate the experimental observation, one need to consider the complex strain at the boundary, which is extremely challenging. Such strain effect makes a bit difference between experimental data and STEM simulations in intensity around the boundary.

Detailed comment 7: “p.8: reference to Fig. S4 should probably be S5”

Response: It is Fig. S4 in the previous version, which give more related data of Figs. 2(M1, M2). Now it has been changed to Fig. S6 in the revised version.

Detailed comment 8: “p.9: “Figs. 3(D2,D2)””

Response: We are sorry for the mistakes. We have corrected it in the revised manuscript.

Detailed comment 8: *“p.10: it is referred to x and y axes, but they are nowhere defined.”*

Response: We have changed it to “*a and b axes*” in the revised manuscript, which are defined in the Figure 4.

REVIEWERS' COMMENTS

Reviewer #1 (Remarks to the Author):

Following the referees' comments, the authors had taken much effort to answer/rebut the questions arising from the previous manuscript. Therefore, I would like to recommend publishing the manuscript in the present form.

Reviewer #2 (Remarks to the Author):

I would like to thank the authors for their response on my report and for addressing the different points in detail, including the realization of additional experiments. I don't have any further comments and thus can recommend the manuscript for publication in Nature Communications.